# Meta-Analysis of Flipped Learning Effects in Nursing Education

**DOI:** 10.3390/ijerph182312814

**Published:** 2021-12-05

**Authors:** Inhee Park, Yeonok Suh

**Affiliations:** 1Department of Nursing, Shinsung University, 1, Daehak-ro, Jeongmi-myeon, Dangjin-si 31801, Chungcheongnam-do, Korea; 2School of Nursing, Soonchunhyang University, 31 Soonchunhyang 6th Rd, Dongnam-gu, Cheonan-si 31151, Chungcheongnam-do, Korea; yeonok@sch.ac.kr

**Keywords:** clinical competence, self-directedness, critical thinking, satisfaction, meta-analysis

## Abstract

Background: This study is a meta-analysis confirming the effect size of clinical competence, critical thinking ability, self-directedness, and learning satisfaction, the outcome variables of flipped learning applied to nursing education. Methods: We selected 18 related studies that analyzed data using CMA (Comprehensive Meta-Analysis 2.2). Results: The effect size of the entire study was Hedges’ g = 0.68 (95% CI = 0.43~0.92). The heterogeneity of the overall effect size was *I*^2^ = 90.7% (Q = 246.67, *p* < 0.001); critical thinking ability had an effect size of Hedges’ g = 0.87, learning satisfaction of Hedges’ g = 0.79, clinical competence of Hedges’ g = 0.53, and self-directedness of Hedges’ g = 0.37. The differences were statistically significant. Conclusion: Flipped learning can effectively improve nursing students’ clinical competence, critical thinking ability, self-direction, and learning satisfaction.

## 1. Introduction

Recently, flipped learning education has been applied to increase the efficiency of education and learning processes due to rapid changes in the educational environment and to convert students into active learners [1,2].

Flipped learning is a learner-centered teaching method in which learners self-learn online before class, asking questions to the instructor and discussing the content with peers during class [3]. This provides a dynamic mutual learning environment for professors and allows students to focus creatively on their learning needs without time and space constraints [4]. In flipped learning, students manage the learning process through learning content before class, so students’ self-discipline and self-regulation can be improved [5]. Additionally, during this class, students can participate in various active learning activities, such as question-and-answer, discussion, group activities, problem-solving, and receiving immediate feedback [6]. A positive change in academic achievement levels has been observed in medical education with an increasing interest in flipped learning classes and reducing boredom [7].

The first study that applied flip learning in the field of nursing education was published in 2013 [8]. A flipped learning model has been applied in nursing education to reinforce nursing competence according to changes in the clinical environment. It is difficult to acquire sufficient knowledge within limited class hours and clinical practice with the existing teaching methods. This is due to the lack of both direct nursing experience in observation-oriented practice and a connection between the theory and clinical practice. Upon employment in a hospital, a gap is noticed in performing practical tasks in clinical nursing [9]. Thus, nursing educators applied flipped learning to improve the quality of education and prepare for complex nursing practice [1,6]. Flipped learning applied to nursing education has a large amount of learning and professional content, and students are inevitably passive in class. It helps students find lecture materials on their own, perform prior learning, and conduct self-directed learning through active learning in class [10]. It analyzes and integrates learning content and is applied to solve complex clinical situations by improving critical thinking skills [11,12]. Additionally, the learner’s perception of flipped learning also shows a positive response, and because they can watch the videos whenever they want, a high level of satisfaction with video lectures during pre-classes is observed [13]. A change in the teaching method of instructors was urged while emphasizing the importance of insight, situational awareness, critical thinking ability, and clinical reasoning to promote learning in nursing students [14].

Various studies have recently been conducted to identify the effectiveness of flipped learning in nursing education. Prior studies have influenced self-directed learning and learner satisfaction [11]. Critical thinking tendencies have significantly increased compared to conventional teaching methods [10]. Additionally, improved academic achievement [15], learning satisfaction, self-efficacy, knowledge, and communication ability [16] indicated positive effects. However, contradictory research results also exist. In clinical practice education, the experimental group that experienced flipped learning improved in clinical performance [11], as compared to the control group that received lecture-type instruction. While some studies outlined no significant results [17], others reported improved critical thinking tendencies [18]. There are limitations in understanding the effectiveness of flipped learning results, including studies in which thinking tendencies are not significant [19]. When the consistency of these individual research results is insufficient, a meta-analysis can be used to identify factors. Regarding the systematic review and meta-analysis of flipped learning applied to nursing education, two foreign studies have focused on the effects of theoretical knowledge and skill scores [20]. In Korea, no meta-analysis has been reported that systematically and quantitatively evaluated the results of flipped learning education. A meta-analysis was conducted because different educational systems and cultural differences in different countries can affect the effectiveness of flipped learning education [21].

Particularly, clinical competence, critical thinking ability, and self-directedness [12], which are the purpose of flipped learning and are important in nursing education, are contributing to the effect of flipped learning education. The application of the flipped learning technique to clinical practice has a positive effect on learning satisfaction [16,22]. Therefore, the purpose of this study was to confirm the effect size on clinical competence, critical thinking ability, self-directedness, and learning satisfaction. Based on these results, this study provides a scientific basis for evaluating the necessity and validity of flipped learning. It is a basis for educational systems and researchers with diverse cultural backgrounds, evaluating the effectiveness of flipped learning models as a teaching–learning method in nursing. 

### Research Aims

This study is a meta-analysis of the effects of flipped learning on educational outcome variables applied to domestic and foreign nursing education, such as clinical competence, critical thinking ability, self-directedness, and learning satisfaction. The specific purpose of this study is as follows:

First, we grasped the general characteristics of flipped learning research selected through the search process. Second, the size of the overall effect of flipped learning applied to nursing students was analyzed. Third, the magnitude of the effect of flipped learning for each outcome variable (clinical competence, critical thinking ability, self-directedness, and learning satisfaction) was analyzed.

## 2. Materials and Methods

### 2.1. Research Design

This study is a meta-analysis study to confirm the effectiveness of clinical competence, critical thinking ability, self-directedness, and learning satisfaction, variables of flipped learning applied to domestic and foreign nursing education. 

### 2.2. Criteria for Literature Selection

We reviewed the Cochrane Handbook for Systematic Reviews of Interventions 5.1.0 [23]. The literature selection criteria were based on the key questions of Populations, Intervention, Comparison, Outcome, Study design (PICOS) per the Preferred Reporting Items for Systematic Reviews and Meta-Analyses (PRISMA) [24]. According to the PICOS, the study subjects (P) were domestic or overseas nursing students, the intervention (I) was flipped learning, comparative mediation (C) conducted for general lecture-style class, the intervention results (O) were related to clinical competence, critical thinking ability, self-directedness, and learning satisfaction and the study types (S) were randomized controlled trials (RCTs) and non-randomized controlled trials (NRCTs). The studies excluded from the literature selection are survey studies, qualitative studies (case studies, descriptive studies), meta-analysis, non-experimental studies, single-group pre- and post-design studies, studies in which the effect size cannot be calculated. In addition, there are studies related to subjects other than nursing students, studies published only as abstracts, and studies in which the language of literature is not English or Korean 

### 2.3. Literature Search Strategy

#### 2.3.1. Literature Search

The literature search in this study was conducted on documents from 1 January 2000 to 30 December 2020, using domestic and international academic search databases (Database, DB). 

When selecting a search center for literature search, the COSI (Core, Standard, Ideal) model suggested by the National Library of Medicine (NLM) is used. Among them, Core is a core part of the literature search, including related domestic literature and core databases (e.g., PubMed, EMBASE, Cochrane central, etc.). In this study, the COre search database recommended by the Korea Institute of Health and Medical Sciences was used. The foreign DBs searched included PubMed, CINAHL, EMBASE, MEDLINE, and Web of Science, and the domestic DBs searched included journals and dissertations published through Research Information Service System (RISS), DBpia, Korean Studies Information Service System (KISS), and KMbase. Additional data search was supplemented with a manual search through the references. The keywords searched included “flip learning”, “flipped learning”, “upside down classroom”, “upside down lesson”, “reverse learning”, “reverse progression lesson”, “effect of flip learning”, “flip classes”, and “nursing students”, “flipped classroom”, “flipped class”, “flipped instruction”, “inverted classroom”, “inverted instruction”, “flipping classroom”, “reverse instruction”, “Students, Nursing [MeSH]”, and “nurs * student *”.

#### 2.3.2. Data Collection and Selection

Data collection was independently searched and reviewed by two researchers, and if they did not agree, they would review the data again, until consensus was reached. A list of papers searched through domestic and foreign DBs was prepared, and duplicate papers were removed using End-Note X9, a bibliographic export program. Reviewing the abstract and full text of the manuscript confirmed whether the study satisfied the research selection criteria according to the data selection criteria. The final analysis literature was determined by excluding articles that did not meet the selection criteria. For the final selected paper, author, publication year, title, study design, study subject, control type, study subject age, subject type, pre-learning method, instruction period, outcome variables, and result values were extracted and recorded in the coding table.

### 2.4. Methodological Quality Assessment

In the methodological quality evaluation of the analysis targets, the RCTs were evaluated for the risk of bias using the Cochrane risk of bias tool (RoB) for randomized trials, and NRCTs were evaluated using the risk of bias tool for non-randomized studies (RoBANs). Two researchers independently conducted these evaluations, and for items that did not match, the study was reviewed together to find consensus. RoB evaluated six areas: generating random assignment order, concealing assignment order, blindfolding research participants and researchers, blindfolding results evaluation, processing insufficient results, and reporting selective results. The RoBAN evaluated eight areas: comparability of target groups, selection of target groups, confounding variables, exposure measurement, blindfolding of outcome assessors, outcome assessment, incomplete outcome data, and selective outcome reporting. Each item is rated as “low”, “uncertain”, or “high” according to the description. 

### 2.5. Data Analysis Method

The data was analyzed using CMA (Comprehensive Meta-Analysis 2.2), a program dedicated for meta-analyses. When calculating the effect size, Cohen’s d tends to overestimate the effect size for small samples. Therefore, Hedges’ g, a corrected standardized mean difference that corrects Cohen’s d, was used [25]. For the interpretation of the effect size, Cohen [26] suggested a small effect size to range between 0.2 and 0.5, a medium effect size to be between 0.5 and 0.8, and a large effect size to be 0.8 or above. The statistical implications of the calculated effect sizes are presented with a 95% confidence interval (CI). The farther away from zero, the stronger the influence. It is interpreted as invalid if the mean effect of zero is included. Recognizing that the research method, group size, intervention method, and comparison group of the studies selected are diverse, a random-effects model that reflected the actual differences between individual studies was applied [26]. In addition, statistical heterogeneity of effect sizes was evaluated using a forest plot, and a funnel plot was used to verify publication bias [27].

### 2.6. Ethical Considerations

The contents and method of this study were IRB review exemption of Soonchunhyang University (IRB-202102-SB-015).

## 3. Results

### 3.1. Selection of Materials

A total of 18 studies were included in the systematic review per the data selection criteria (illustrated in Figure 1). In the first stage, 681 articles were searched using the search strategy for each database. The overlapping 314 episodes were excluded using the bibliographic management program. In the second step, 135 papers were selected by reviewing the titles and abstracts of 232 papers. In the third stage, the original texts of the 135 selected papers were reviewed, and the criteria for exclusion of the research subject and research design were applied. Of these, four non-original papers, one non-English paper, 27 non-conforming research results, six non-nursing students, and 59 studies where the research design did not match the criteria were excluded. Twenty studies were selected from 115 studies. In the fourth stage, the full texts of the 20 studies were reviewed. Two of them were excluded because the result value could not be extracted due to insufficient statistical data, and the final 18 studies were confirmed for meta-analysis.

### 3.2. Characteristics of Documents Subject to Systematic Literature Review 

The research design of the 18 papers selected for the analysis of this study was NRCT 88.9% and RCT 11.1%, and the publication type was journal 88.9% and thesis 11.1%. As for the subject type applied to the intervention, practical subjects accounted for the most at 50%, and video (video content, video learning, lecture video) as the pre-education method was 72.2%. The sample size range of the experimental group was 22 to 287, the control group was 21 to 198, and the number of interventions was 4 to 14. The program application time was 2–3 h per session, but there were studies that did not describe the intervention method in detail. The proportion of outcome variables was 38.9% for clinical performance, 27.8% for critical thinking ability, 27.8% for self-direction, and 33.3% for learning satisfaction (Table 1).

### 3.3. Evaluation of Literature Quality

Considering the results of the 18 methodological quality assessments in this study, 50% of the studies had a low RoB in randomization order in two RCT papers. For the concealment items in the order of assignment, the low RoB was 50%, and the items for blindfolding of researchers had 100% uncertain RoB. In the case of blindfolding and selective reporting of results, the low RoB was 100%, and the high RoB in insufficient result data was 100%. In the 16 NRCT papers, the low bias was 100% in the subject group comparability, exposure measurement, result evaluation, and selective result reporting. In selecting a target group, the low RoB was 50%, the high RoB was 37.5%, and the uncertainty of the RoB was 12.5%. Regarding the confounding variables, the low RoB was 81.25%, and the high RoB was 18.75%. In the evaluator’s blindfold, the low RoB was 25%, and the uncertainty RoB was 75%. The unsafe result data showed that the low RoB was 25%, and the high RoB was 68.75%; the uncertainty of the RoB was 6.25%. A single group was excluded from the literature selection criteria, and papers with clear pre- and post-evaluations and those planned were included in the literature containing all expected results. The quality of the papers was found to be higher than average because of the overall evaluation (Table 1)

### 3.4. Effect Size of Flipped Learning

Table 2 and the forest plots present the effect size of flipped learning applied to nursing education on clinical competence, critical thinking ability, learning satisfaction, and self-directedness (Figure 2). The effect size of the entire study was Hedges’ g = 0.68 (95% CI = 0.43~0.92), showing a statistically significant median effect size.

The heterogeneity of the total effect size was *I*^2^ = 90.7% (Q = 246.67, *p* < 0.001), appearing to fall within a large range [18]. The effect size of each effect variable of the flipped learning intervention was statistically significant because the 95% CI excluded 0, and the critical thinking ability was Hedges’ g = 0.87, showing the largest effect size. Next, learning satisfaction (Hedges’ g = 0.79), clinical competence (Hedges’ g = 0.53), and self-directedness (Hedges’ g = 0.37) are shown in Table 2. 

By verifying the intervention effect, the subject type and the prior learning method type were used as modulators. In the subject method type, the effect size value is calculated by the practical subjects, the practical and theoretical integrated subjects, and the theoretical subjects. In the practical subjects had Hedges’ g = 0.95, the practical and theoretical integrated subjects had Hedges’ g = 0.36, and the theoretical subjects had Hedges’ g = 0.30. According to the subject type, there was a significant difference in the flipped learning effect (Q = 13.59, *p* = < 0.001). In the prior learning method type, as the effect size value is calculated by classifying the video method and the method where the video and lecture book were given simultaneously, the Hedges’ g was 0.71 for the video method and 0.45 for the video and lecture book; however, there was no statistically significant difference (Q = 1.22, *p* = 0.269).

### 3.5. Publication Bias 

The publication bias of this study was analyzed by Egger’s regression test. As a result of statistically checking the significance of the degree of asymmetry, it was found that there was no publication bias (t = 0.82, *p* = 0.42) (Figure 3).

## 4. Discussion

This study systematically examined and meta-analyzed the effects of flipped learning on clinical competence, critical thinking ability, self-directedness, and learning satisfaction applied to nursing education to present the effects in an integrated and objective manner. Therefore, the main findings are as follows:

The selected 18 studies considered flipped learning as the intervention group and traditional lecture-style learning as the control group. Of these, the two (11.1%) randomized controlled pre- and post-design studies using an RCT design were conducted overseas. Sixteen pieces (88.9%) were pre-post designs for the non-equivalent control group using the NRCT design, where six were conducted overseas and ten were conducted domestically.

Considering the items with high RoB or uncertain RoB in the methodological quality evaluation of theses, seven papers (38%) in selection bias were not randomly assigned. Part 2 (12.5%) did not present a specific method for generating the randomization order, and in Part 3 (16.6%), the order of allocation was not concealed. Thus, the researcher could predict the intervention group. Additionally, in terms of performance bias, no specific method for blinding research participants and researchers was presented in 14 papers (77.7%). In flipped learning, as instructor intervention is required, there are limitations to blindfolding participants and researchers, influencing the meta-analysis results. Despite the limitations in RCT research methods in flipped learning, various approaches to verify the quality of learning methods must be sought.

Regarding the subject types that applied flipped learning, practical subjects were the highest, with nine studies (50%), followed by seven theoretical subjects (38.9%) and two theory and practical subjects (11.1%). The theoretical subjects were critical thinking, nursing process, pathophysiology, health assessment, and patient safety, and the practical subjects were simulation practice, basic nursing practice, and clinical nursing practice.

Among the analysis papers, when applying flipped learning in practical subjects, nursing techniques cannot be directly performed. Thus, the methods and procedures of nursing techniques are repeatedly learned through audio-visual data [11]. Video content was used [19] or developed and applied by the instructor [31]. As the learning experience provided by the medium is different, the instructors set a clear goal through flipped learning, and the curiosity and motivation of learners were met.

As a result of the meta-analysis of flipped learning studies applied to 18 studies, the overall average effect size was 0.68, corresponding to the median effect size. This is similar to the study showing the median effect size of 0.58 as the overall average effect size of the learning effects of domestic flipped learning [40] and the study showing a median effect size of 0.59 as the effect size of flipped learning on domestic college students [41]. It was confirmed that flipped learning was more effective than traditional lecture-style classes when applied to nursing education.

As a result of comparing the effect sizes of the flipped learning outcome variables in this study, they were found in this order of critical thinking ability, learning satisfaction, clinical competence, and self-direction.

The critical thinking ability had a Hedges’ g of 0.87, showing a large effect size and a high level of heterogeneity. In five studies, the experimental group to which flipped learning was applied showed improved critical thinking ability compared to the control group. It can be predicted that flipped learning, which allows students to ask questions and solve problems on their own through prior learning and to improve critical thinking skills through discussion with other students during class in the classroom [42], is effective in nursing education. Additionally, a previous study [19] showed that although it is possible to acquire knowledge and skills, there is a limit to critical thinking. Instructors must grasp the learner’s disposition, attitude, and learning strategy and find and search for information rather than memorize knowledge; based on this, it is necessary to develop a running program.

As a result of analyzing the effect of flipped learning on learning satisfaction, the intermediate effect size was found to be Hedges’ g = 0.79. When learning flipped learning, information was provided to students, such as videos, animations, and images of clinical cases, inducing interest and resulting in positive responses. Satisfaction was high, as this allowed flexibility to learn at a desired speed and time [43,44]. However, there have also been studies with low learning satisfaction due to tasks assigned from prior learning during flipped learning, online environmental problems for learning, and adaptation to new teaching methods [8]. The instructor should consider the learner’s learning environment when developing flipped learning activities. It is necessary to consider avoiding a learning gap, such as understanding the infrastructure for watching videos and prior learning, and it is necessary to analyze the learning conditions in detail per the learning goal. In other words, it is necessary to promote understanding of learning by providing students with a plan for the learning content and learning method of flipped learning in advance so that they can familiarize themselves with the content to be learned. Additionally, the learner’s learning process was constantly monitored. A strategy for linking the content of prior learning and offline activities is needed by providing feedback on prior learning in the classroom.

As a result of analyzing whether flipped learning is more effective than traditional methods in improving nursing students’ clinical nursing skills, the effect size of clinical competence had a Hedges’ g of 0.53, showing a medium-sized effect. This confirmed that learning the procedure and techniques through videos was effective [45], and it was also confirmed by another study [46] that the clinical performance was improved by flipped learning in clinical practice. Nursing is a practical study. Direct nursing skills and repetitive practice are required to acquire and improve such skills. However, it is difficult to obtain qualitative improvement because of the inability to perform direct skills during flipped learning. When flipped learning is applied in practical classes, nursing skills and nursing theories are learned based on evidence, and in classroom sessions, the contents of prior learning can be directly performed. Additionally, it effectively improves students’ nursing skills through immediate feedback from instructors [47]. Factors for achieving clinical performance include interpersonal relationships and communication, nursing process, critical thinking, nursing intervention, creative thinking, basic nursing, and problem-solving ability [48]. These are improved by applying flipped learning to the subject in the corresponding domain, helping them to effectively perform the clinical nursing skills corresponding to the detailed elements.

In this study, the effect size of self-directedness was small (Hedges’ g = 0.37). The most important aspect of flipped learning is to increase the degree of self-directedness, and in many research results, self-directedness was improved through flipped learning [49,50]. Learners watched video lectures on pre-learning to familiarize themselves with the learning content, and the activities in the classroom were focused on learners. Notably, the self-directedness is improved due to the increased sense of responsibility and activeness in all learning processes [51]. Self-directedness is an important variable affecting learning outcomes, such as learning motivation, academic achievement, and satisfaction [52]. Through flipped learning, the instructors foster the learner’s learning ability and act as facilitators such that learners can have flexibility and develop lifelong learning habits for self-directedness [53]. Accordingly, when the learner’s characteristics and teaching methods are matched, self-directedness is achieved successfully. When designing a flipped learning curriculum, it is necessary to develop educational methods to improve self-directedness ability by focusing on learner-centered education. 

Other variables related to the subject type and pre-learning method were the modulating variables that explained the difference in effect size. As a result of meta-ANOVA using the subject type as the moderator variable, the following were practical subjects, theory and practice mixed design subjects, and theory subjects.These results showed a significant difference in the flipped learning effect according to the subject type (Q = 13.59, *p* ≤ 0.001), and it was confirmed that the effect of a flipped learning class was higher in practical subjects. In a practical subject, flipped learning helps one experience knowledge integrated with observation and experience [12]. The interaction between the instructor and students increases, and the nursing skills gradually improve [8]. Studies have confirmed this effect. Additionally, the effect size value was calculated by classifying the learning methods into video method (15 episodes) and the method where video and lecture books were given (five episodes) simultaneously, indicating that the learning by the video method was higher but not statistically significant (Q = 1.22, *p* = 0.269). This could be because when the video and lecture book were given simultaneously, the number of assignments to be learned increased, consequently increasing the learning burden, and the lack of time and grade were not reflected. It is necessary for the instructor to explain in advance how the pre-learning videos are related to the lecture contents in order for the students to plan an appropriate amount of pre-study.

It was difficult to measure various variables because there were differences in the specific research methods such as the flip learning education method applied to each study, the number of applications, and the time. In future research, it is necessary to improve the quality of research by describing specific research methods for teaching and learning designs applied to flipped learning classes, such as the characteristics of instructors and learners and the learning process.

Critical thinking, clinical reasoning, and clinical performance are required for nursing students to adapt to the rapidly changing nursing environment [14]. Applying flipped learning to equip self-directedness ability, students learn how to acquire abilities and competencies by applying various information rather than simply accumulating knowledge and facing problems in a medical nursing environment.

The limitations of this study are as follows. An extensive search on flipped learning was performed using flipped learning keywords and a series of other relevant keywords. However, the possibility of publication bias cannot be excluded because the publication languages are limited to English and Korean.

This study selected RCT and NRCT research among the studies related to flipped learning applied to nursing education to increase the reliability of the research results. The academic significance included confirming the magnitude of the effect on clinical Competence, critical thinking ability, self-directedness, and learning satisfaction as outcome variables, an effect that was determined to exist. 

## 5. Conclusions

Flipped learning applied to nursing education is a learning method that effectively improves nursing students’ critical thinking ability, learning satisfaction, clinical competence, and self-directedness. In particular, it helps increase critical thinking ability and clinical competence necessary in nursing education, which occupies a large part of clinical practice education. Several studies have confirmed that flipped learning is being applied to practical education. However, the research results on students’ prior learning methods, the time required, and educator preparation are insufficient. Therefore, detailed methods for future learning types should be developed, and research to prove their effectiveness should be conducted.

Therefore, it is recommended that flipped learning education be actively used as an effective nursing education teaching method. It is expected that continuous research on the effects and learning effects on students will be conducted.

## Figures and Tables

**Figure 1 ijerph-18-12814-f001:**
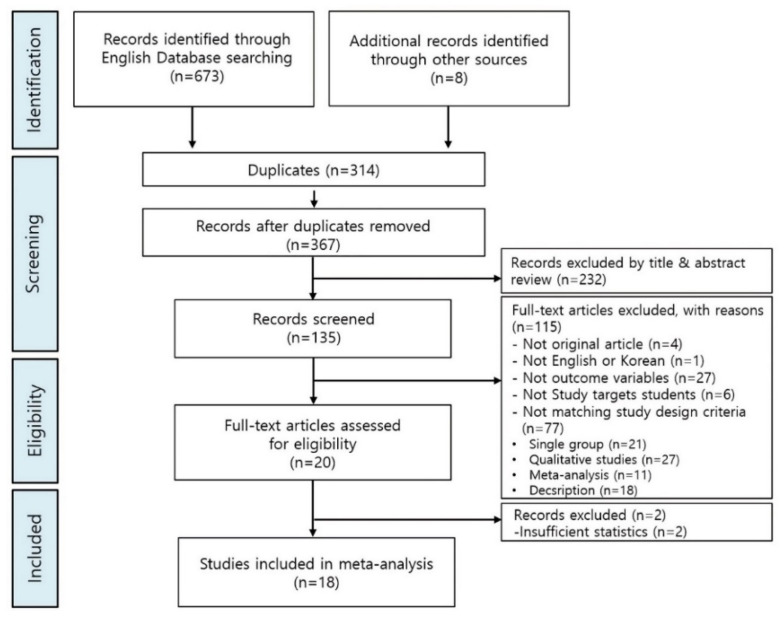
Flowchart of study selection.

**Figure 2 ijerph-18-12814-f002:**
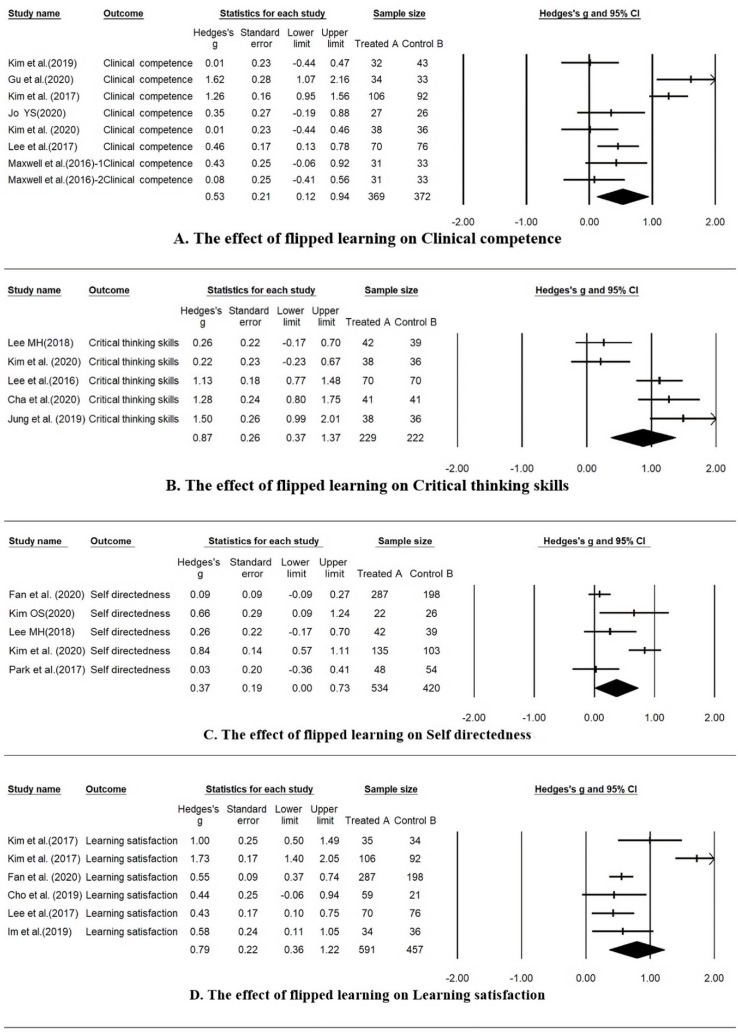
Forest plot of effect size of flip learning for nursing students.

**Figure 3 ijerph-18-12814-f003:**
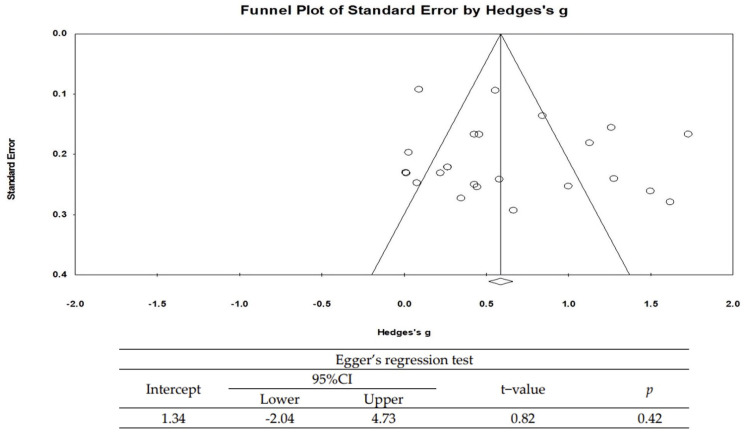
Publication bias.

**Table 1 ijerph-18-12814-t001:** Descriptive summary of included studies.

No	Author (Year)	Publication	Control Group Program	Participants		ST	Subject	Outcomes	Moderator Variables				Methodological Quality Assessment									
				Sample Size					Subject Type		Pre-Class Activities		SD	RCT	-	R	AC	BP	BO	-	IO	SR
				Exp.	Con.				Theory	Practice	Video	Document		NRCT	CT	SP	CV	MI	BO	ER	IO	SR
1	Cha et al. (2020) [18]	Journal	TAU	41	41	6	Critical Thinking and Nursing Process	Critical thinking skills	○		+	+	NRCT		①	①	ⓗ	①	ⓤ	①	ⓗ	①
2	Fan et al. (2020) [28]	Journal	TAU	287	198	-	Theory practice	Self-directedness Learning satisfaction	○	○	+	+	NRCT		①	①	ⓗ	①	ⓤ	①	ⓗ	①
3	Gu et al. (2020) [29]	Journal	TAU	34	33	-	Simulation practice	Clinical competence		○	+		NRCT		①	ⓗ	①	①	①	①	①	①
4	Kim et al. (2020) [19]	Journal	TAU	135	103	5	Critical Thinking and Nursing Process	Self-directedness	○		+		NRCT		①	ⓗ	①	①	①	①	①	①
5	Kim et al. (2020) [30]	Journal	TAU	38	36	4	Basic Nursing Practice	Clinical competence Critical thinking skills		○	+		NRCT		①	①	①	①	ⓤ	①	①	①
6	Kim OS. (2020) [31]	Journal	TAU	22	26	10	Basic Nursing Practice	Self-directedness		○	+		NRCT		①	ⓗ	①	①	ⓤ	①	ⓗ	①
7	Jo YS. (2020) [32]	Thesis	TAU	27	26	4	Emergency Nursing	Clinical competence	○		+	+	NRCT		①	①	①	①	ⓤ	①	①	①
8	Cho et al. (2019) [11]	Journal	TAU	59	21	-	Clinical adult nursing practicum	Learning satisfaction		○	+	+	NRCT		①	①	①	①	ⓤ	①	ⓗ	①
9	Jung et al. (2019) [33]	Journal	TAU	38	36	-	Basic Nursing Practice	Critical thinking skills		○	+		NRCT		①	①	①	①	ⓤ	①	①	①
10	Kim et al. (2019) [34]	Journal	TAU	32	43	14	Patient safety course	Clinical competence	○		+		NRCT		①	ⓗ	①	①	①	①	①	①
11	Im et al. (2019) [22]	Journal	TAU	34	36	6	Psychiatric Nursing Practice	Learning satisfaction		○	+		NRCT		①	ⓤ	①	①	ⓤ	①	①	①
12	Lee MH. (2018) [35]	Thesis	TAU	42	39	5	Pathophysiology	Critical thinking skills Self-directedness	○		+		NRCT		①	①	①	①	ⓤ	①	①	①
13	Kim et al. (2017) [36]	Journal	TAU	106	92	8	Simulated learning	Clinical competence Learning satisfaction		○	+		RCT			ⓗ	ⓤ	ⓤ	①		ⓗ	①
14	Kim et al. (2017) [16]	Journal	TAU	35	34	-	Clinical practice	Learning satisfaction		○	+		RCT			①	①	ⓤ	①		ⓗ	①
15	Lee et al. (2017) [37]	Journal	TAU	70	76	6	Psychiatric Nursing Practice	Clinical competence Learning satisfaction		○	+		NRCT		①	ⓗ	①	①	①	①	①	①
16	Park et al. (2017) [38]	Journal	TAU	48	54	9	Basic nursing course	Self-directedness	○		+		NRCT		①	ⓗ	①	①	ⓤ	①	①	①
17	Lee et al. (2016) [39]	Journal	TAU	70	70	6	Health assessment	Critical thinking skills	○	○	+		NRCT		①	①	①	①	ⓤ	①	①	①
18	Maxwell et al. (2016) [17]	Journal	TAU	31	33	-	Patient safety	Clinical competence	○		+	+	NRCT		①	ⓤ	ⓗ	①	ⓤ	①	ⓤ	①

TAU = Treatment as Usual; Exp. = Experimental group; Con. = Control group; ST = Session Total; SD = Study design; RCT = Randomized Controlled Trial; NRCT = Randomized Controlled Clinical Trial; Video; R = Random sequence generation; AC = allocation concealment; BP = Blinding of participants and personnel; BO = Blinding of outcome assessment; IO = Incomplete outcome date; SR = Selective reporting; CT = Comparability of target group; SP = Selection of participant; CV = Confounding variables; MI = Measurement of intervention (exposure); ER = Evaluation of results; ① = low risk of bias; ⓗ = high risk of bias; ⓤ = uncertain risk.

**Table 2 ijerph-18-12814-t002:** Effects of outcome and moderator variables.

Categories			k	Hedges’ g	95% CI		Heterogeneity		
					Lower	Upper	*Q*	*p*	I^2^%
Total effect size			24	0.68	0.43	0.92	246.67	<0.001	90.7
Outcome variables		Clinical competence	8	0.53	0.12	0.94	50.86	<0.001	86
		Critical thinking skills	5	0.87	0.37	1.37	26.04	<0.001	85
		Learning satisfaction	6	0.79	0.36	1.22	45.53	<0.001	89
		Self-directedness	5	0.37	0.00	0.73	24.48	<0.001	84
Moderator variables	ST	Practice	12	0.95	0.71	1.20	13.59	<0.001	
		Theory	9	0.30	0.01	0.58			
		Theory and Practice	3	0.36	0.08	0.80			
	PCA	Video	17	0.71	0.46	0.95	1.22	0.269	
		Video and document	7	0.45	0.08	0.83			

k = number of effect size; 95% CI = 95% confidence interval; Q = total variability; I^2^ = between-study variability; ST = Subject type; PCA = Pre-class activities.

## Data Availability

Data from the study are available upon request.

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
