# Peer review of "Meta-Analysis of Flipped Learning Effects in Nursing Education"

_ijerph, 2021, doi:10.3390/ijerph182312814_

Round 1

Reviewer 1 Report

Firstly, I would like to thank the author(s) for the submission of this paper. The focus of this paper is interesting and significant to the field of higher education by exploring the effect of a student-centered approach, as Flipped Learning (FL), on the preparation of nurses.  However, the paper in its current form requires additional work.

1. Be consistent along with all the paper about the aim of the study – abstract (.. examining the effect size of clinical competence, ….), introduction (… to confirm the effect size of learning satisfaction), and research aims (meta-analysis of the effects of flipped learning on educational outcome…).

2. In the Introduction some concepts are used but not clarified (e.g. what are traditional instructional online and offline classes?); Who tells that FL was used in nursing education since 2007? (Bergman and Sam? Right?). This section would also benefit from a more depth of understanding of the benefits of FL and a more in deep connexion between FL and active learning.

3. Concerning material and methods:

  • Why these databases? Why the inclusion of dissertations on the domestic databases? What criteria for the manual search? It’s not enough said the keywords used;
  • Concerning the range of this review, the search was made up to December 30, 2020, since when?;
  • The exclusion criteria must be presented, not only refer “that the criteria of exclusion of the research subject and research design were applied”;
  • Why include randomized controlled trials (RCT) and non-randomized controlled trials (NRCT) studies? At the end of the discussion is stated that the inclusion of RCT and NRCT studies increases the reliability of the research results. I have some concerns about this option because only 2 studies are RCTs and given that RCTs are the most stringent way of determining whether a cause-effect relation exists between one intervention and the outcome.
  • Why include studies that don't describe the intervention methods in detail?

4. Results and discussion

  • I missed descriptive data from the studies. The results would benefit from an inclusion of a content description of each group of studies, using for example the variables explored (critical thinking, …) or the categories identified, as the kind of pre-class work. It isn’t enough saying in 5 studies the experimental group (….); it is necessary to identify the studies.
  • The discussion section benefited if the data are not mixed – data from RCT (only 2 studies) and NFCT studies, and data from different pre-class activities, among other categorizations, needs different interpretations.

Finally, the conclusion can be improved if the results include a section with more detail from each group of studies, for example, the ones that used pre-classes DVD.

Author Response

Thank you for reviewing my dissertation.

Reviewer 2 Report

This well-written paper puts numbers on the effectiveness of flipped learning effects in nursing by doing a literature review of 18 papers that address this topic.  I would like to know more about the dynamic mutual learning environment for professors, particularly because the material prepared in advance is likely to be reused.  As a professor, flipped learning is a lot more fun and the dialogue with students more exciting than just lectures.  I would have liked to see some comments from professors who were using this method.  I am curious as to where the date 2007 came from.  I think this method has been used well before that date, although the terchnology has inproved significantly over recent years.  I was confused by the material in lines 71-78.  Why would flipped learning affect satisfaction rather than  clinical thinking ability?  Thoughts as to why self-directedness was so low?

Define acronyms when first used RCT and NRCT.  I like the included discussion of the analytics you used.  In Table 2, k vs K.  

Problems with discussion.  Delete Therefore line 255. Line 256-257. not 18 nursing students.  Confusing 0.67 in text and 0.68 in table.  The rest of the discussion was excellent.

Author Response

Thank you reviewer for your opinion.

Point 1: This well-written paper puts numbers on the effectiveness of flipped learning effects in nursing by doing a literature review of 18 papers that address this topic.  I would like to know more about the dynamic mutual learning environment for professors, particularly because the material prepared in advance is likely to be reused.  As a professor, flipped learning is a lot more fun and the dialogue with students more exciting than just lectures.  I would have liked to see some comments from professors who were using this method.I am curious as to where the date 2007 came from.  I think this method has been used well before that date, although the terchnology has inproved significantly over recent years.  I was confused by the material in lines 71-78.  Why would flipped learning affect satisfaction rather than  clinical thinking ability?  Thoughts as to why self-directedness was so low?

Define acronyms when first used RCT and NRCT.  I like the included discussion of the analytics you used.  In Table 2, k vs K.  

Problems with discussion.  Delete Therefore line 255. Line 256-257. not 18 nursing students.  Confusing 0.67 in text and 0.68 in table.  The rest of the discussion was excellent.

1. I would have liked to see some comments from professors who were using this method.I am curious as to where the date 2007 came from. 

Response 1: The content has been changed.

The first study that applied flip learning in the field of nursing education was published in 2013[8]. A flipped learning model has been applied in nursing education to reinforce nursing competence according to changes in the clinical environment.

2. Define acronyms when first used RCT and NRCT.  I like the included discussion of the analytics you used.  In Table 2, k vs K.  

Response 2: Changed to 'k'.

*k=number of effect size

3. Problems with discussion.  Delete Therefore line 255. Line 256-257. not 18 nursing students.  Confusing 0.67 in text and 0.68 in table.  The rest of the discussion was excellent.

Response 3: As a result of the meta-analysis of flipped learning studies applied to 18 studies, the overall average effect size was 0.68, corresponding to the median effect size.

Round 2

Reviewer 1 Report

Firstly, I would like to thank the author(s) for the effort put into the revision process.

Most of the recommendation was attended and the explanation given in some doubts about some options are adequate, but few recommendations aren't solved:

  1. Concerning the aim, I suggest deciding if is 'examine' (abstract) or 'confirm' (introduction and research aims).
  2. Concerning material and methods: The databases selection needs to be better explained. Why are they better for this thematic?
  3. The conclusion can be improved with more detail from each group of studies, for example, the ones that used pre-classes DVD.

Author Response

Thank you reviewer for your opinion.

Point 1:

Concerning the aim, I suggest deciding if is 'examine' (abstract) or 'confirm' (introduction and research aims).

Response 1:

abstract : line 10

I changed it.

examine--> confirm

Point 2:  Concerning material and methods: The databases selection needs to be better explained. Why are they better for this thematic?

Response 2:

Changed line 122

When selecting a search center for literature search, the COSI (Core, Standard, Ideal) model suggested by the National Library of Medicine (NLM) is used. Among them, Core is a core part of literature search, including related domestic literature, core databases (e.g., PubMed, EMBASE, Cochrane central, etc.). In this study, the COre search database recommended by the Korea Institute of Health and Medical Sciences was used.

Point 3: The conclusion can be improved with more detail from each group of studies, for example, the ones that used pre-classes DVD.

Response 3:

line 202 add

video(video content, video learning, lecture video) as the pre-education

Change the word.

DVD --> video